# Pre-Impact Detection Algorithm to Identify Tripping Events Using Wearable Sensors

**DOI:** 10.3390/s19173713

**Published:** 2019-08-27

**Authors:** Federica Aprigliano, Silvestro Micera, Vito Monaco

**Affiliations:** 1The BioRobotics Institute, Scuola Superiore Sant’Anna, 56127 Pisa, Italy; 2Bertarelli Foundation Chair in Translational Neuroengineering, Center for Neuroprosthetics and Institute of Bioengineering, School of Engineering, Ecole Polytechnique Federale de Lausanne, 1015 Lausanne, Switzerland; 3IRCCS Fondazione Don Carlo Gnocchi, 20148 Milan, Italy

**Keywords:** pre-impact detection, tripping, wearable sensors, lower-limb biomechanics

## Abstract

This study aimed to investigate the performance of an updated version of our pre-impact detection algorithm parsing out the output of a set of Inertial Measurement Units (IMUs) placed on lower limbs and designed to recognize signs of lack of balance due to tripping. Eight young subjects were asked to manage tripping events while walking on a treadmill. An adaptive threshold-based algorithm, relying on a pool of adaptive oscillators, was tuned to identify abrupt kinematics modifications during tripping. Inputs of the algorithm were the elevation angles of lower limb segments, as estimated by IMUs located on thighs, shanks and feet. The results showed that the proposed algorithm can identify a lack of balance in about 0.37 ± 0.11 s after the onset of the perturbation, with a low percentage of false alarms (<10%), by using only data related to the perturbed shank. The proposed algorithm can hence be considered a multi-purpose tool to identify different perturbations (i.e., slippage and tripping). In this respect, it can be implemented for different wearable applications (e.g., smart garments or wearable robots) and adopted during daily life activities to enable on-demand injury prevention systems prior to fall impacts.

## 1. Introduction

Falling is widely recognized as one of the most important causes of disability in fragile individuals [1,2,3,4]. Several reports agree with the evidence that about 30% of older adults (+65 years of age) fall at least once per year [5,6], and the percentage increases with ageing and related neuro-musculo-skeletal diseases [7]. Falls can indeed result in traumatic and physiological consequences [3], thus worsening the quality of life of fragile individuals and augmenting the costs of healthcare [8,9,10]. The increasing life expectancy of the worldwide population is expected to further exacerbate the effects of the risk of falls on society as a whole. As such, national and international agencies have been facing this problem by consistently supporting research activities in the field of fall prevention programs [11,12].

One of the strategies currently being investigated to counteract the risk of falls involves predicting the forthcoming occurrence using wearable sensors in conjunction with suitable signal processing algorithms [13,14,15]. Inertial sensors, also named inertial measurement units (IMUs; i.e., accelerometers and/or gyroscopes), are among the most applicable sensor types since they can work as stand-alone platforms during daily activities in unstructured environments and can be easily embedded in either garments or wearable devices [13,16,17].

Several authors have developed and tested different fall-risk assessment tools, such as machine learning approaches to classify fall risk (e.g., high vs. low) based on the analysis of signals recorded by IMUs during specific assessment tasks (e.g., walking, time up-and-go, sit-to-stand transitions) [3,18,19]. This approach is frequently used in clinical settings since it allows the stratification of potential fallers in terms of prospective fall risk probability. Nonetheless, they do not distinguish people based on their attitude to manage hazards met during daily activities (e.g., slippery surfaces, carpets). Accordingly, this approach cannot be used to predict a loss of balance that suddenly escalates to a fall.

Additionally, others have investigated the effectiveness of a closed-loop strategy combining predictive algorithms that are able to detect signs of incipient falls—the pre-impact fall detection algorithm—with wearable devices designed to counteract the lack of balance. In particular, once the lack of balance is detected, the algorithm employs a wearable device designed to either mitigate the impact with the ground [20,21,22] or supply suitable assistance [23,24], thus preventing potential injuries.

We recently developed and tested a novel pre-impact algorithm to detect the lack of balance due to unexpected slippages delivered while steadily walking in well-controlled experimental conditions [25]. This algorithm is based on the evidence that a sudden perturbation (i.e., slippage) alters the quasi-periodic features of steady walking, thus highlighting an abnormal behavior that can reflect a lack of balance and, in turn, an incipient fall (see Section 2.3). The previous version of our algorithm was designed to be easily implemented in an active pelvis orthosis; that is, a wearable robot inherently equipped with joint position sensors. Accordingly, it parses out the measured angles at hip joints and enables an assistive strategy mediated by the active pelvis orthosis [23,24]. It is worth noting that the same approach, after being suitably re-tuned, is expected to be able to detect unexpected perturbations by analyzing different types of signals as long as they reflect the rhythmic features of steady walking. In this respect, it is possible to hypothesize that our algorithm is still effective if processing signals are transmitted from inertial sensors.

In this study, we tested the effectiveness of our pre-impact detection algorithm (PIDA) while identifying the lack of balance following an unexpected tripping incident. Specifically, the orientation of lower limb segments was estimated by processing the output data of wearable IMUs (see Section 2 for further details). These data were used as inputs for our PIDA, which was tuned to minimize both detection time and false alarms (FA). Finally, the best location of the sensor and best performance were identified.

## 2. Materials and Methods

### 2.1. Participants, Experimental Setup, and Protocol

Eight healthy young volunteers (6 females and 2 males; 25.9 ± 2.8 years; 58.4 ± 6.3 kg; 1.67 ± 0.09 m) were enrolled for this study. All subjects signed an informed consent before starting the experimental sessions.

Participants were asked to manage sudden and unexpected tripping perturbations delivered while they were steadily walking at 1 m/s on a treadmill. For safety purposes, the treadmill was equipped with handrails to allow participants to grasp them only in case of an unrecoverable lack of balance. Otherwise, they were not allowed to use the handrails during the entire experimental session.

Perturbations were provided by a custom mechatronic platform designed to host and break a nylon rope (7 mm diameter) through a cam-based mechanism (Figure 1). Particularly, the cam was driven by a DC motor (2342S024CR) with a planetary gear head (30:1), an incremental encoder (HEDS 5500A), and a related driver (MCDC3006S), compounded and provided by Faulhaber Minimotor SR, Croglio, Switzerland. The nylon rope was connected on one side to the main frame of the mechatronic platform via a compliant spring (stiffness 3 N/m) and on the other side to one of the participant’s feet. During steady walking, the rope moved forward or backward, using a series of pulleys, according to the foot movement. If enabled, the cam stopped the movement of the rope and inhibited the foot from moving forward along the swing phase, emulating tripping. To prevent gait asymmetries, the mechatronic platform was equipped with two rope-spring systems, even though the connection for the unperturbed foot was not equipped with cam-braking mechanism.

The control architecture of the platform was based on an Arduino Due microcontroller and managed two different inputs: (1) an external enabler, handled by the experimenter; and (2) a foot switch under the unperturbed foot signaling its heel strike. When the experimenter decided to deliver the perturbation, she/he enabled the control loop using an external input. Then, the cam-based actuation started moving when the heel strike of the unperturbed (i.e., left) foot was detected. The movement of the rope was then stopped for 0.9 s and then released to allow the subject to autonomously recover balance. To prevent anticipative behaviors, participants could not see the experimenter and listened to music with headphones during the whole experimental session.

The experimental protocol consisted of five unexpected tripping perturbations delivered during the swing phase of the right leg and ten additional trials, in which no perturbation was applied. To prevent bias, subjects did not know whether they would have been perturbed or not, and they did not know the side being perturbed.

Seven IMUs (Xsens wireless Motion Tracker Awinda system; [26]) were used as motion trackers. Each IMU had embedded inertial sensor components, namely a 3D rate gyroscope, a 3D accelerometer, a 3D magnetometer, a barometer, and a thermometer. They were located on lower limbs (i.e., pelvis, thighs, shanks and feet) to monitor the elevation angles of each segment, with a sampling rate of 100 Hz.

The 3D kinematic of the feet was also recorded by a 6-camera based Vicon 512 Motion Analysis System (Vicon, Oxford, UK), with a sampling rate of 100 Hz, to identify gait events and record the onset of the perturbation (i.e., heel strike of the left foot).

Kinematic data and inertial signals were synchronized off-line. In more detail, before each trial, participants were asked to perform a couple of jumps from a steady stance while both the camera-based system and IMUs were recording body kinematics. Datasets were off-line time-aligned by assessing the time-lag between them. Specifically, we computed the cross-correlation of the second time-derivative of the vertical component of one marker on the IMU placed on the right foot with the vertical component of the acceleration detected by the same IMU within a 5-s time window including the two initial jumps. The time-lag between records coincided with the abscissa of the maximum of their related cross-correlation function.

Research procedures followed the Declaration of Helsinki and were approved by the Local Ethical Committee.

### 2.2. Data Pre-Processing

Gait events (i.e., right and left heel strikes) were identified based on the trajectories of the markers placed on the feet, as reported in the literature [27].

For each subject and each trial, data were subdivided in two subsets: data recorded before and after the onset of the perturbation (i.e., left heel strike). The former referred to the last 10 unperturbed strides in which each cycle started with the left heel strike and ended with the following one; each stride was time-interpolated over 101 points, and all strides were averaged to have a representative unperturbed gait cycle. The latter referred to the compensatory stride; it started simultaneously with the onset of the perturbation and ended with the following left heel strike. Data referring to the compensatory stride were also time-interpolated over 101 points.

The duration of both strides (i.e., unperturbed and compensatory ones) was evaluated as the interval between two consecutive heel contacts of the left foot.

Lower limbs (i.e., perturbed and unperturbed limbs; PL and UL, respectively) were modeled as a three-link (i.e., thigh, shank and foot) chain. According to the aim of this study, we primarily monitored the elevation angles in the sagittal plane—that is, the orientation of right thigh, shank and foot with respect to the vertical axes (see red lines in Figure 1); in particular, the elevation angles were estimated using a new Kalman filter specifically developed by Xsens for capturing human motion by fusing the gyroscope, accelerometer and magnetometer signals [26].

For each trial, an 18-s long dataset—i.e., 15 s before the onset of the perturbation and 3 s after—was retained for the tuning and the validation of the detection algorithm.

### 2.3. Pre-Impact Detection Algorithm (PIDA)

Our PIDA was designed to signal sudden modifications of the quasi-periodic features of walking patterns due to unexpected perturbations [25]. To achieve this task, it accounted for two main components (Figure 2): (1) a set of adaptive oscillators (AOs) coupled with a kernel based non-linear filter, and (2) an adaptive threshold-based algorithm (ATBA).

The set of adaptive oscillators (AOs), if properly tuned, provided a synchronized estimation of non-sinusoidal quasi-periodic input signals with zero phase-lag [28]. Accordingly, the input and output of this predictor are likely to be similar during steady walking. Conversely, if a perturbation suddenly altered the cyclic features of the gait patterns, the output seeks a new periodic signal, thus diverging from the actual input. The accuracy and the responsiveness of the pool of AOs were optimized by tuning their learning gains, namely phase (*k_P_*) and amplitude (*k_A_*), learning gains.

The difference between the input and output of the AOs—i.e., the error—was then analyzed by the ATBA (Figure 2), as follows:The algorithm first selected a *w*-long portion of the error signal prior to the current time frame and computed its mean (*μ*) and standard deviation (*σ*);Then, it compared the absolute value of the error signal at the current time-frame with a threshold set at *μ* + *kσ*, where *k* represents a corrective factor to shape the value of the threshold;If the absolute value of the error was above the threshold, the algorithm delivered a warning;A set of *r* consecutive warnings was used to detect a lack of balance that could potentially result in an incipient fall.

The optimization of our PIDA consisted of identifying a suitable set of tuning parameters—i.e., *k_P_*, *k_A_*, *w*, *k* and *r*—to both minimize the detection time and reduce the number of FA.

### 2.4. PIDA Tuning

The tuning of our PIDA consisted of identifying an optimal set of tuning parameters—i.e., *k_P_*, *k_A_*, *w*, *k* and *r*—to both minimize the detection time (i.e., the time elapsing between the onset of the perturbation and the output of the algorithm) and reduce the number of FA; i.e., false positives (FP) and false negatives (FN). Noticeably, an FP occurred when a postural transition was detected before the onset of the actual perturbation, and an FN occurred when no postural transitions were detected within 1 s after the onset of the perturbation.

To achieve the best tuning of our PIDA, we first investigated the dynamic behavior of the AOs in the domain of learning gains (i.e., *k_A_* and *k_P_*) reported in Table 1. Specifically, we sought the domain of learning gains that allowed for the best match between the real and estimated elevation angles of the right shank and foot. To achieve this task, we computed the root means square of the difference (RMSD) and the Pearson correlation coefficient (*ρ*) between current and estimated angles during the 3-s long time window before the onset of the perturbation. Accordingly, we assumed that a suitable tuning of the AOs was obtained if the RMSD was lower than 0.1 rad and *ρ* was higher than 0.9.

Once the AOs were set up, we tuned the ATBA (Figure 2) to identify the best performance of our PIDA while parsing out datasets collected during the experimental sessions. The ranges of the tuning parameters for the ATBA (i.e., *w*, *k* and *r*) are reported in Table 1. The performance of our PIDA was assessed in terms of mean detection time (MDT) and percentage of FA across subjects and tripping trials.

## 3. Results

### 3.1. Time Course of Elevation Angles

Figure 3 shows the lower limb behavior during the unperturbed and compensatory strides. Before the onset of the perturbation, the elevation angles were comparable to those described in the literature at the same speed [29,30]. As expected, after the onset of the perturbation, lower limb kinematics, especially those that referred to the perturbed shank and foot, were altered due to the cable braking effect (Figure 3a,c,e). Accordingly, before the onset of the perturbation (i.e., during steady walking), the unperturbed stride lasted 1.25 ± 0.06 s; after the onset of the perturbation (i.e., during the tripping event), the compensatory stride lasted 1.09 ± 0.09 s.

### 3.2. Tuning of the AOs

Considering that tripping mainly modified the orientation of the perturbed shank and foot (Figure 3), the performances of our algorithm were investigated while monitoring these segments (see also Section 4). Accordingly, Figure 4 shows representative examples of accurate and inaccurate tuning of the AOs tracking the orientation of the perturbed shank and foot.

Figure 5 shows the results of the tuning of the AOs and reports the range of learning gains (i.e., *k_A_* and *k_P_*) allowing for a suitable prediction of the elevation angles during steady walking. As far as the elevation angle of the perturbed shank is concerned, a subdomain of *k_A_* and *k_P_* (range: *k_A_* = 1 and *k_P_* = [10, 20, 30, 40, 50, 60, 70, 80, 90, 100]) allowed the pool of AOs to properly fit real kinematics (i.e., RMSD < 0.1 rad, and *ρ* > 0.9; Figure 5a,b). Conversely, only few learning gains of the AOs (i.e., *k_A_* = 1 and *k_P_* = [80, 90, 100]) allowed for a suitable tracking of the perturbed foot elevation angle (Figure 5c,d). As expected, if the AOs were properly tuned, the error signal between the output of the AOs and the real kinematics suddenly increased after the onset of the perturbation (Figure 4). This abrupt change of the error signals was then analyzed by the ATBA to detect the lack of balance due to tripping events.

### 3.3. Tuning of the ATBA

Based on the results of the AOs’ tuning, the detection algorithm (i.e., ATBA) was validated on a subdomain of tuning parameters (*k_P_*, *k_A_*, *w*, *k*, and *r*). Specifically, the learning gains of the AOs allowing for the best tracking of the input signals were selected as follows: for the perturbed shank, *k_P_* and *k_A_* were set at 20 and 1, respectively (Figure 4a); for the perturbed foot, *k_P_* and *k_A_* were set at 100 and 1, respectively (Figure 4c).

Figure 6 shows the MDT (panel a) and the percentage of FA (panel b) obtained by monitoring the elevation angle of the perturbed shank for all tuning parameters of the ATBA (i.e., *w*, *k* and *r*). The results revealed that, for a suitable FA percentage (i.e., lower than 10%), the best combination of tuning parameters was *w* = 400, *r* = [6, 8] and *k* = 3.5. In particular:If *r* = 6, the MDT was 0.37±0.11 s, with FA equal to 9.4%;If *r* = 8, the MDT was 0.40±0.15 s, with FA equal to 9.4%.

As far as the elevation angle of the perturbed foot is concerned, the percentage of FA was 100% for all the combinations of tuning parameters of the ATBA (i.e., *w*, *k* and *r*). Noticeably, these FA were all FN; that is, the ATBA was never able to detect signs for the lack of balance within a 1-s time window following the onset of the perturbation. Accordingly, no MDT was obtained for all these conditions.

## 4. Discussion

The aim of this study was to investigate the performance of a pre-impact detection algorithm while identifying unexpected tripping disturbances delivered during steady walking. To do this, a previous version of our algorithm, relying on joint angles and identifying slippages [25], was updated to detect the lack of balance following tripping perturbations. Noticeably, in this study, wearable sensors were used to monitor the orientation of lower limb segments.

Overall, the best performance of the pre-impact detection algorithm was obtained monitoring the orientation of the perturbed shank, achieving an MDT equal to 0.37 ± 0.11 s with an acceptable rate of FA (lower than 10%). Noticeably, the time course of the thigh elevation angle was not significantly altered by the perturbation (Figure 3) and one of our recent studies revealed that if our PIDA parses out hip joint angles during tripping, the detection time increases to 800–900 ms [31]. Therefore, we decided to avoid a deeper analysis of the performance of our PIDA while parsing out the elevation angle of the most distal lower limb segment, in accordance with the purpose of this study (i.e., to find out the best sensor-set to minimize the detection time).

This result, in conjunction with our previous findings [23,25], corroborates the hypothesis that the proposed algorithm can detect the lack of balance due to different acute perturbations (i.e., slippage, tripping) delivered during steady walking in a timely manner. Specifically, with respect to this study, our algorithm is well suited to be implemented in a smart lower limb prosthesis, equipped with IMUs, to enable strategies to promote balance recovery.

### 4.1. Tuning of the AOs

Firstly, the tuning of amplitude and phase learning gains of the pool of AOs (i.e., *k_A_* and *k_P_*) were effectively updated to calculate the error signal (i.e., the input of the ATBA). Remarkably, according to the aim of this study, during walking trials, the error signal should be around zero to avoid FP. Conversely, during tripping trials, the error signal should increase to avoid FN and identify the lack of balance.

As shown in Figure 5, the behavior of the AOs, while tracking the shank orientation, was acceptable (RMSD < 0.1 rad, and *ρ* < 0.9) in a wider subdomain of learning gains (i.e., *k_A_* and *k_P_*; Figure 5a,b) compared to that allowing for an accurate foot tracking (Figure 5c,d).

This result was expected since it reflects the behavior of the AOs to better track signals with a lower frequency content. In fact, the AOs behave like low-pass filters with zero delay [32]. Noticeably, the frequency content of the shank orientation was lower than that of the foot (Figure 7); thus, the AOs can more accurately track the kinematics of the proximal body segments during steady locomotion. Accordingly, before the onset of the perturbation, the error signal achieved while monitoring the orientation of the shank (Figure 4a) was lower than that observed while monitoring the orientation of the foot (Figure 4c). In any case, the percentage of FP was minimized in both cases (i.e., monitoring the orientation of both shank and foot).

After the onset of the perturbation, the error signal achieved while monitoring the orientation of the shank increased (Figure 4a), allowing a fast identification of the balance loss (Figure 6a). Conversely, after the perturbation onset, the error signal observed while monitoring the orientation of the foot was close to zero (Figure 4c). This latter result was likely due to the fact that the AOs were tuned to track a signal with higher frequency content (i.e., foot orientation), which were similar to those elicited while people reactively managed unexpected perturbations. In other words, the AOs tuned to monitor the orientation of the foot were also able to suitably track the higher frequency content resulting from sudden and unexpected tripping disturbances. In fact, no significant differences were observed between the estimated and measured signals after the perturbation (Figure 4c); thus, our approach relying on an IMU placed on the foot did not distinguish the steady walking from the reactive responses elicited after tripping.

It is worth noting that increasing the domain of learning gains above the range reported in Table 1 did not improve the performance of the AOs, as it only potentially increased the above-mentioned effect (data not reported).

Overall, a trade-off between FP and FN should be advantageous to allow the ATBA to provide a suitable MDT. Future analyses will be focused on improving the performances of the AOs in terms of FA.

### 4.2. Tuning of the ATBA

The tuning of the ATBA consisted of properly selecting the length of the bin before the current sample to gather some statistical properties of the signal (i.e., *w*), the number of consecutive warnings to prevent FA (i.e., *r*) and the threshold amplitude (i.e., *k*).

Concerning the first parameter (i.e., *w*), the results revealed that acceptable values of FA (<10%) can be obtained with *w* = 400 samples (corresponding to 4 s); that is, a time window roughly accounting for 4 full strides, as in our experimental conditions. Noticeably, *w* = 400 samples also represent the minimum time required to update the ATBA at the beginning of each session. Based on the evidence that about 60% of daily bouts account for more than 5 full strides [33], the tuning of our ATBA relying on *w* = 400 samples is supposed to be suitable for the majority of daily activities. A greater *w* would reduce the initial responsiveness of our algorithm and require a greater memory mostly due to storage-related reasons. Accordingly, we believe that *w* = 400 samples represent a suitable trade-off between the prompt tuning of the algorithm and a low effort in term of data management and storage.

With respect to the number of warnings (i.e., *r*), we had to choose a suitable value to guarantee a fast detection time and prevent the risk of FA. In this study, acceptable values of FA (<10%) were achieved with *r* = 6 and *r* = 8 samples. Noticeably, the increase in *r* induced a delay in the detection of the lack of balance. Thus, the best performance, in terms of both minimized detection time and FA, was obtained using *r* = 6 samples (see Figure 6).

The last parameter of our algorithm was the corrective factor to shape the threshold (i.e., *k*). In our previous study [25], we heuristically chose *k* = 3 based on the assumption that the distribution of the error signal was Gaussian; thus, the probability that a value is over three standard deviations is lower than 1%. However, we acknowledged that a different choice might slightly improve the algorithm performances. Accordingly, in the current study, we tested three values (3, 3.5 and 4), showing that the best performance involved *k* = 3.5 obtaining a threshold equal to *μ* ± 3.5*σ*.

Overall, according to the reported results, we can conclude that the proposed pre-impact detection algorithm, if well-tuned, can be effective across different scenarios (e.g., slippages, tripping events) showing suitable values for detection time and FA.

### 4.3. Sensors Position and Related Performance of the Algorithm

The previous version of our algorithm, relying on hip joints kinematics [25], was designed to be easily implemented in an active pelvis orthosis, a wearable robot equipped with joint position sensors, and to detect the lack of balance due to unexpected slippages delivered during steady walking. A later analysis confirmed that the proposed strategy was effective in closing a human–robot loop aimed at promoting balance recovery after slippages in elderly people and trans-femoral amputees [23,24].

Here, we proposed an updated version of this algorithm relying on wearable sensors placed on the lower limb segments. In particular, according to the purpose of our study, we tested the algorithm considering only the distal segments (i.e., shank and foot), since they were more affected by the tripping perturbations (see Figure 3). In addition, we only used signals recorded from the perturbed side (i.e., the right one) as the input of our algorithm, considering that it is earlier and more significantly altered by the perturbation (Figure 3c,e) compared to the unperturbed one (Figure 3d,f).

Overall, the best performance was achieved tracking the orientation of the perturbed shank; thus, the proposed pre-impact detection algorithm can be effectively implemented by using only one IMU placed on this body segment.

It is important to highlight that the advancement in micro-technology and wireless communication makes wearable sensors suitable for pre-impact detection algorithms [34]. Indeed, wearable IMUs are a low-cost system that can be used to detect fall in extended spaces, and they do not require additional infrastructure installation. Accordingly, our pre-impact detection algorithm could potentially be part of a complete fall detection and injury prevention system that would promote more independent living in the elderly community.

### 4.4. Comparison with the State of Art

Over the last few years, a great deal of effort has been put into investigating new fall detection strategies to automatically identify the occurrence of a fall event [20,35,36,37]. Fall detection systems can be generally classified as post-fall mobility detection and pre-impact detection [34]. The former is expected to provide timely medical assistance for fall victims. However, falls can be only detected after impacts; thus, related injuries cannot be prevented. The latter, as with our approach, is expected to overcome such limitations, allowing falls to be detected before the body hits the ground. Accordingly, this strategy also has the potential to enable on-demand fall protection systems to prevent fall-related injuries. Thus, the main advantage of the pre-impact fall detection is that, if a fall can be detected in its earliest stage, more efficient preventive systems can be implemented for the minimization of injuries. Noticeably, our pre-impact detection algorithm can identify a lack of balance due to unexpected tripping events in about 0.37 ± 0.11 s (Figure 6a).

We must acknowledge that this outcome cannot be directly compared to those reported in the literature. As a matter of fact, other authors determined either the critical falling time (i.e., the time which elapsed from the fall detection time and the moment at which the inclination angle between the center of mass and the center of pressure exceeded a range of −23° to 23° from the vertical [38]) or the lead-time (i.e., the time which elapsed from the fall detection and the impact of the subjects on a mattress [20,39,40,41]). In contrast, in this study, we investigated the time which elapsed from the onset of the perturbation and the actual detection; that is, the time window preceding that observed by the above-mentioned works. However, our outcomes (i.e., MDT = 0.37 ± 0.11 s; Figure 6a), in conjunction with the evidence that the duration of the transitory phase between steady locomotion and hitting the ground can be longer than 0.5–0.7 s [29,42,43], allow us to hypothesize that the proposed pre-impact detection algorithm is able to promptly signal a lack of balance due to tripping, providing enough time to effectively enable mitigation strategies for impact prevention.

Although promising, the performances of our detection algorithm are limited by the fact that it was tested and validated under well-controlled experimental conditions. In this respect, the cyclic features of the unperturbed gait patterns were guaranteed by a constant walking speed, in contrast to the evidence that human behavior, in real life, is much more variable (e.g., walking speed changes continuously, while walking people can change direction, or climb/descend stairs [33]). In addition, the lack of balance was induced by pseudo-impulsive events (i.e., an unexpected tripping), whereas a real fall is a complex motor task involving no-stereotyped biomechanics [44]. Finally, we used the simplest approach (i.e., a threshold-based algorithm parsing only one signal) to detect abnormal behaviors even if a more complex strategy can be also implemented to improve the overall performance. Accordingly, future analysis will be focused on testing and updating the proposed algorithm to detect real-world falls.

## 5. Conclusions

In this study, a pre-impact detection algorithm was updated to identify the lack of balance due to tripping. The best performance was obtained when analyzing the perturbed shank with a mean detection time equal to 0.37 ± 0.11 s and a low rate of false alarms (<10%). To conclude, the proposed algorithm is a simple threshold-based approach for the automatic detection of different types of unexpected gait disturbances [25], monitoring the shank orientation with a wearable sensor. Accordingly, it can be easily implemented in a lower limb robotic prosthesis already equipped with sensors to provide assistance to the user while regaining balance after unexpected tripping.

## Figures and Tables

**Figure 1 sensors-19-03713-f001:**
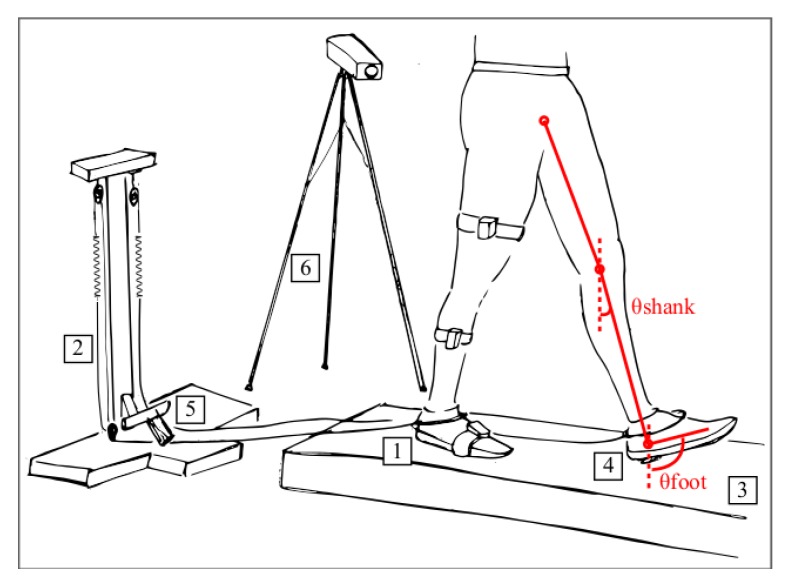
Schematic of the experimental setup. Labeled components are as follows: 1. perturbed foot; 2. spring-rope mechanism; 3. treadmill; 4. footswitch under the unperturbed foot; 5. cam-based braking mechanism; 6. camera-based motion capture system. As a representative example, the position of the Inertial Measurement Units (IMUs) is reported on the thigh, shank and foot of the right limb, and the elevation angles used as input of the algorithm (i.e., elevation angles of the shank and foot—θshank and θfoot, respectively) are depicted on the left limb.

**Figure 2 sensors-19-03713-f002:**
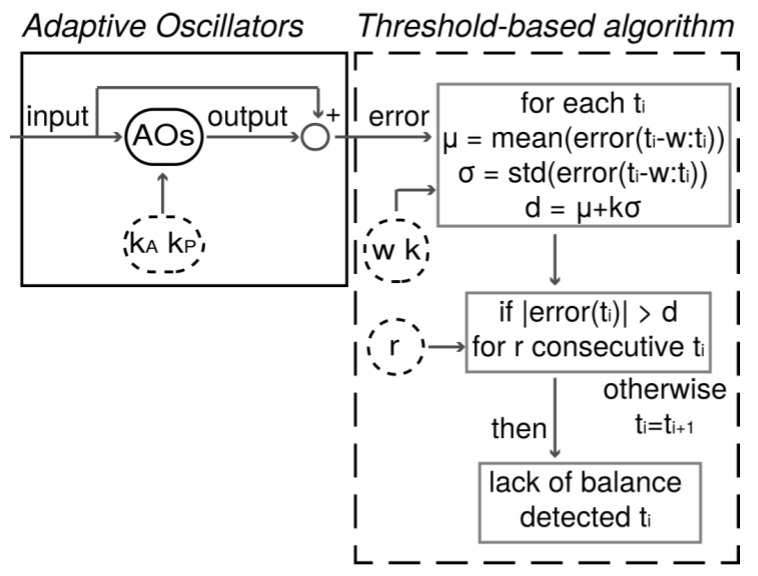
Pre-impact detection algorithm (PIDA) composed of a set of adaptive oscillators (AOs) (box on the left) and the adaptive threshold-based algorithm (dashed box on the right). Tuning parameters (i.e., *k_A_*, *k_P_*, *w*, *k* and *r*) are depicted in dashed ovals.

**Figure 3 sensors-19-03713-f003:**
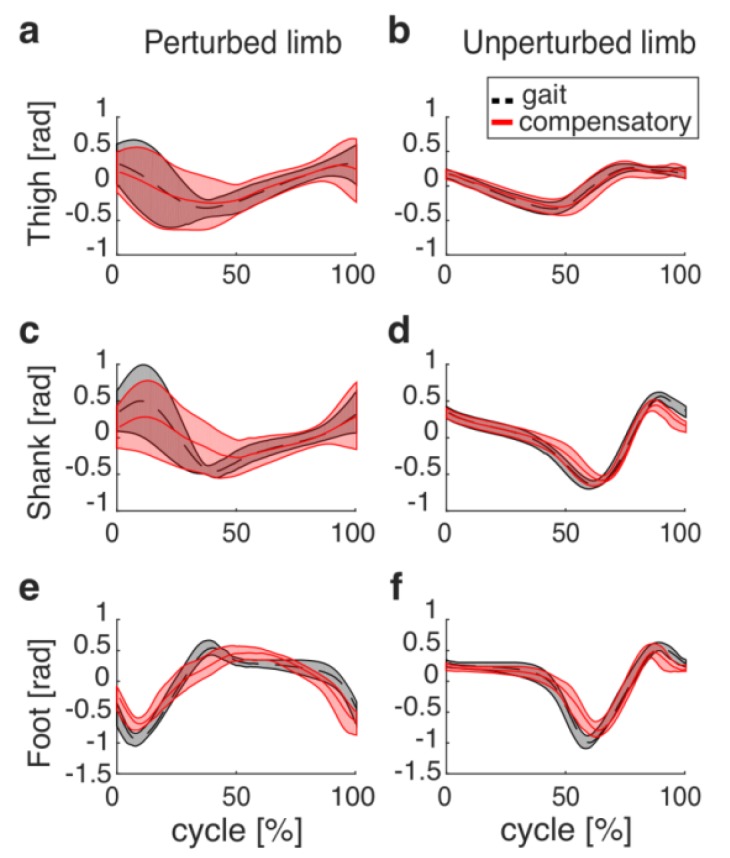
Elevation angles of the thigh, shank and foot recorded before (dashed black lines) and after (red lines) the perturbation onset. *x*-axes represent the time course (in %) of the gait and compensatory cycles; 0 and 100% correspond to two consecutive heel strikes of the left foot. Mean (bold) and one standard deviation (shaded areas) are shown for both the perturbed ((**a**), (**c**) and (**e**)) and unperturbed limb ((**b**), (**d**) and (**f**)) segments.

**Figure 4 sensors-19-03713-f004:**
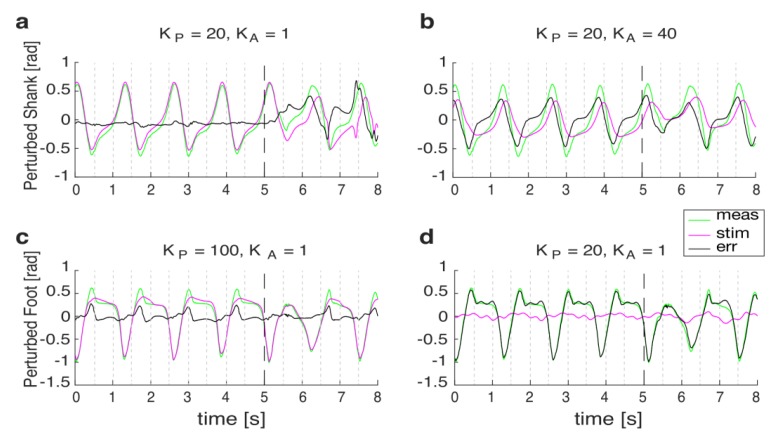
Representative examples of the AOs’ tuning, showing measured signals by the IMUs (green lines), the estimated signals by the AOs (magenta lines) and their differences (i.e., error; black lines). Accurate and inaccurate tunings of the AOs are reported for the perturbed shank and foot ((**a**), (**b**), (**c**) and (**d**), respectively) before and after the onset of the perturbation (i.e., the dashed lines at 5 s).

**Figure 5 sensors-19-03713-f005:**
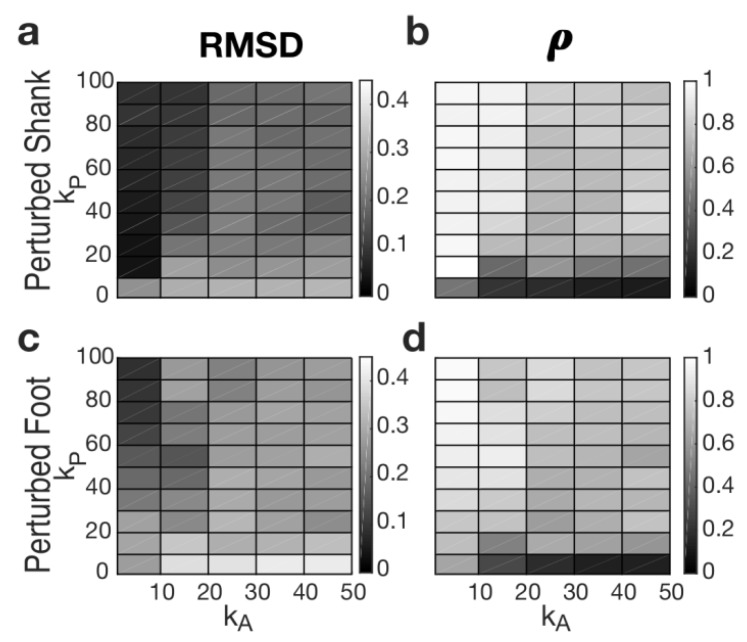
RMSD ((**a**) and (**c**)) and *ρ* ((**b**) and (**d**)) for each learning gain of the AOs (i.e., *k_A_* and *k_P_*) and for both input signals (i.e., elevation angles of perturbed shank and foot). *x*-axes show k_A_; *y*-axes show *k_P_*. Dark and light gray describe low and high values, respectively.

**Figure 6 sensors-19-03713-f006:**
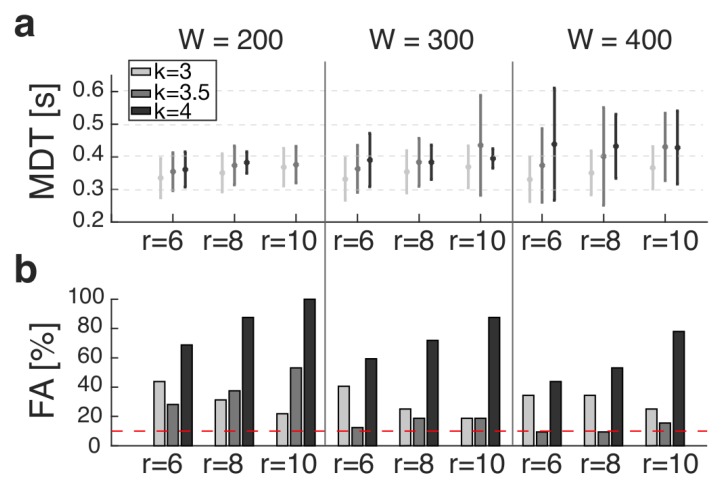
(**a**) Mean detection time (MDT) and (**b**) percentage of false alarms (FA) across all subjects and tripping trials. Light gray, gray and black lines represent values for *k* = [3, 3.5, 4], respectively. In panel (**b**), the dashed horizontal line represents an FA equal to 10%.

**Figure 7 sensors-19-03713-f007:**
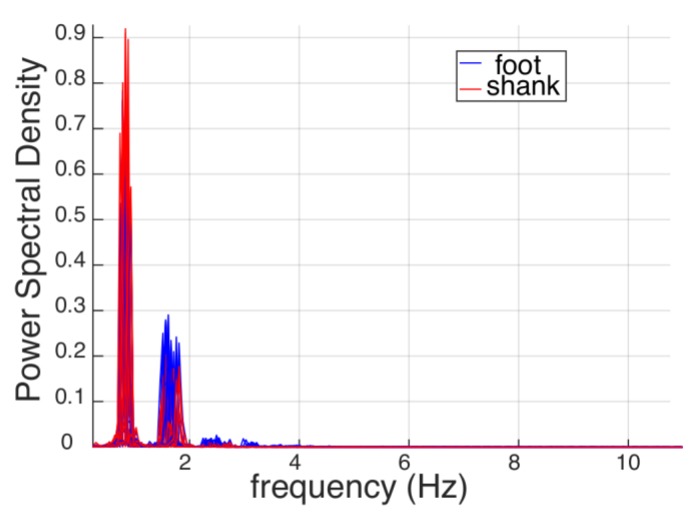
Power spectral density of the elevation angles of the perturbed foot (blue lines) and perturbed shank (red lines). Signals of each subject and each trial are superimposed.

**Table 1 sensors-19-03713-t001:** Tuning parameters.

Tuning Parameter	Description	Values
*k_P_*	Phase learning gain of the adaptive oscillators	From 1 to 100, step 10
*k_A_*	Amplitude learning gain of the adaptive oscillators	From 1 to 50, step 10
*w*	Length of the observed bin	200, 300, 400 samples
*k*	Shaping factor of the threshold	3, 3.5, 4
*r*	Maximum number of warnings	6, 8, 10

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
