# Peer review of "Pre-Impact Detection Algorithm to Identify Tripping Events Using Wearable Sensors"

_sensors, 2019, doi:10.3390/s19173713_

Round 1

Reviewer 1 Report

spelling error in line: 43, 46, 47, 49, 52, 55, 57, 59, 70, 84, 93, 132, 138, 139, 147,  151, 158, 164, 167 (2x), 199, 200, 222 (3x), 223, 224, 225, 227(2x), 228, 232, 243, 244, 247, 249, 250, 272, 274, 283, 285, 293, 294, 295, 297, 299, 302, 304(2x), 305, 308, 311, 313, 319, 323, 329, 339(2x), 346(2x), 348, 351, 354, 362, 363, 365(2x), 366, 367, 369, 370, 376, 378, 379, 384, 389, 394, 397, 405, 408,

You are talking about 8 volunteers but provide only the relevant data for the 6 females (at leats it looks like that).

Parenthesis opened in line 114 is not closed.

Is the sampling rate equal to 100 Hz or is it 100Hz?

How was the synchronisation done between the Vicon system and the IMU data?

Reviewer 2 Report

I think research is acceptable. I will make following suggestions

1) An actual photograph of the experimental should be provided with the article.

2) "Thermometer was used as motion tracker?" I could not understand and it may need explanation.

3) Section 2-3 states of optimization used to find kp, ka, w, ksand r to minimize detection time? I will like the equation of detection time and algorithm to minimize it. The equation and algorithm must precede this statement. 

Reviewer 3 Report

The manuscript is well written and presented with some interesting results in identifying tripping events using wearable sensors. The methodology, results and discussion are done very scientifically and highly acceptable. only concern is that the wearable sensors are meant to be on the body, dress or shoes without any external physical connections.  In the experiments presented n this manuscript, the authors have used external pulley systems connected by rope to the foots. Though this could be acceptable for lab-scale experiments, authors should find alternative methods to sense the required movements with sensors on real wearables.

Authors need to address this issue in the conclusions as the future work.

Reviewer 4 Report

This manuscript presents a pre-impact detection algorithm, in which an adaptive threshold-based algorithm, relying on a pool of Adaptive Oscillator, is tuned to identify abrupt kinematics modifications during tripping. The input for the algorithm is the elevation angle of the perturbed shank, which is measured by an IMU. Besides, the effect of the parameters in the algorithm on the detection time and false alarms is studied so as to achieve good performance. In general, the novelty and contribution of this manuscript are ok. However, the following issues should be addressed before accepting:

1. The authors should carefully check the texts in the manuscript. Many words are interconnected.

2. “optimal/ optimized” shouldn’t be used in the manuscript, because no optimization algorithm is used to optimize the parameters in the system. The authors just compare the performance when the parameters are set to a set of values.

3. The authors just consider the elevation angles of shank and foot. How about the performance when the input is the elevation angle of the thigh?  It can be measured by an IMU mounted on the thigh. Thus, I think the authors should add this part in the revised version.

Round 2

Reviewer 4 Report

The authors have answered my questions, and the current version can be accepted.